# Effects of Tolvaptan on Oxidative Stress in ADPKD: A Molecular Biological Approach

**DOI:** 10.3390/jcm11020402

**Published:** 2022-01-13

**Authors:** Matteo Rigato, Gianni Carraro, Irene Cirella, Silvia Dian, Valentina Di Vico, Lucia Federica Stefanelli, Verdiana Ravarotto, Giovanni Bertoldi, Federico Nalesso, Lorenzo A. Calò

**Affiliations:** Department of Medicine, Nephrology, Dialysis and Transplantation Unit, University of Padova, 35128 Padova, Italy; matteo.rigato@hotmail.it (M.R.); carraro.gianni@gmail.com (G.C.); irene.cirella@gmail.com (I.C.); silvia.dian@gmail.com (S.D.); valentina.divico@gmail.com (V.D.V.); luciafederica.stefanelli@aopd.veneto.it (L.F.S.); verdiana.ravarotto@aopd.veneto.it (V.R.); giovanni.bertoldi@studenti.unipd.it (G.B.); federico.nalesso@unipd.it (F.N.)

**Keywords:** ADPKD, oxidative stress, tolvaptan

## Abstract

Autosomal dominant polycystic disease (ADPKD) is the most frequent monogenic kidney disease. It causes progressive renal failure, endothelial dysfunction, and hypertension, all of which are strictly linked to oxidative stress (OxSt). Treatment with tolvaptan is known to slow the renal deterioration rate, but not all the molecular mechanisms involved in this effect are well-established. We evaluated the OxSt state in untreated ADPKD patients compared to that in tolvaptan-treated ADPKD patients and healthy subjects. OxSt was assessed in nine patients for each group in terms of mononuclear cell p22^phox^ protein expression, NADPH oxidase key subunit, MYPT-1 phosphorylation state, marker of Rho kinase activity (Western blot) and heme oxygenase (HO)-1, induced and protective against OxSt (ELISA). p22^phox^ protein expression was higher in untreated ADPKD patients compared to treated patients and controls: 1.42 ± 0.11 vs. 0.86 ± 0.15 d.u., *p* = 0.015, vs. 0.53 ± 0.11 d.u., *p* < 0.001, respectively. The same was observed for phosphorylated MYPT-1: 0.96 ± 0.28 vs. 0.68 ± 0.09 d.u., *p* = 0.013 and vs. 0.47 ± 0.13 d.u., *p* < 0.001, respectively, while the HO-1 expression of untreated patients was significantly lower compared to that of treated patients and controls: 5.33 ± 3.34 vs. 2.08 ± 0.79 ng/mL, *p* = 0.012, vs. 1.97 ± 1.22 ng/mL, *p* = 0.012, respectively. Tolvaptan-treated ADPKD patients have reduced OxSt levels compared to untreated patients. This effect may contribute to the slowing of renal function loss observed with tolvaptan treatment.

## 1. Introduction

Autosomal dominant polycystic kidney disease (ADPKD) is a genetic multisystem disorder with an autosomal dominant pattern and an estimated prevalence of affected subjects between 1:400 and 1:1000. It is characterized by multiple, bilateral renal cysts and is found in 10% of patients with end-stage renal disease (ESRD) [1].

ADPKD is caused by mutations in the PKD1 (78% of cases) or PKD2 (15% of cases) genes. PKD1, located on chromosome 16 (16p13.3), encodes polycystin-1 (PC1), a large multidomain glycoprotein that is cleaved at a proteolytic site of the G protein-coupled-receptor. PKD2, located on chromosome 4 (4q21), encodes for polycystin-2 (PC2), a protein belonging to the transient receptor potential family of calcium-regulated cation channels.

Both PC1 and PC2 are found on primary cilia and studies now support an inhibitory role of PC1 and PC2 against cystogenesis [2]. Loss of PC1 or PC2 is associated with low intracellular calcium concentrations, which lead to increased activity of adenyl cyclase type 5 and 6, reduced activity of phosphodiesterase 1, excessive concentrations of cyclic AMP (cAMP), and consequent cystogenesis through the activation of the proliferation and secretion pathways [2]. 

Preclinical studies showed the role of arginine-vasopressin-mediated cAMP signaling as a driver of cystic proliferation and fluid secretion in ADPKD. In mouse models, the suppression of vasopressin release, vasopressin V2 receptor antagonism, or genetic elimination of vasopressin resulted in improved cyst burden [3].

Tolvaptan, a vasopressin V2 receptor antagonist, was shown to be effective in slowing the increase in total renal volume during a 3-year study period and it is now a consolidated treatment option for ADPKD patients [4].

Oxidative stress (OxSt), inflammation and endothelial dysfunction are non-traditional cardiovascular risk factors that have a key role in the induction of atherogenesis, malnutrition, and anemia in patients with chronic kidney disease (CKD). OxSt is one of the main factors leading to the progression of CKD [5,6]. Increased OxSt in patients with CKD has been linked to a reduction in antioxidants and reduced bioavailability of nitric oxide (NO) [7,8,9]. The RhoA/Rho kinase (ROCK) system is deeply involved in a broad spectrum of essential physiological processes, including having a key role in the induction of OxSt signaling [5,10,11]; the activation of the RhoA/Rho kinase pathway essentially leads to cardiovascular-renal remodeling via the induction of OxSt and the reduction in NO bioavailability [5,10,11,12].

While the impact of OxSt and the effects of tolvaptan on OxSt in ADPKD have been evaluated in vitro and in animal models via the determination of biochemical markers related to OxSt and inflammation [13,14], no human studies are available. To this end, using a molecular biology approach, we evaluated ex vivo the OxSt status of ADPKD patients in our cohort treated with tolvaptan in terms of (1) mononuclear protein expression of p22^phox^, a subunit of NADPH oxidase key in the transfer of electrons to molecular oxygen to generate superoxide anions and the induction of OxSt [15]; (2) phosphorylation state of MYPT-1, a marker of RhoA/ROCK pathway activity [10,12]; and (3) HO-1 levels, induced and protective against OxSt [16], and compared these to a group of matched untreated ADPKD patients with normal renal function and to a group of healthy subjects.

## 2. Patients and Methods

We enrolled in this study 9 patients of our cohort of ADPKD patients, 5 women and 4 men, mean age 46 ± 5 years, treated with tolvaptan, followed at the Polycystic Kidney Outpatient Clinic of the Nephrology, Dialysis and Transplant Unit at Padova University Hospital). Additionally, 9 untreated ADPKD patients (7 women and 2 men, mean age of 34 ± 11 years) with normal renal function (eGFR > 80 mL/min/1.73 m^2^), and 9 healthy subjects (5 men and 4 women, mean age 31 ± 5 years) were used as control groups.

To minimize the possibility that different quantitative protein expressions between mononuclear cell subtypes could influence the gene expression of the OxSt-related proteins we considered in our study, i.e., through a fluctuation in the number of different mononuclear cell subtypes, we checked the patients for the absence of changes in biochemical markers of inflammation such as CRP, α2 globulin, and monocyte and lymphocyte counts, as well as for no clinical evidence of infectious or inflammatory disease before the beginning of the study. In addition, all the study participants, including the healthy subjects, were non-smokers, and all the subjects abstained from food, alcohol, coffee, and caffeine-containing beverages for at least 12 h prior to the blood sampling.

Informed consent was obtained from all the study participants, who were informed of the nature of the study, which required only a blood sample obtained at the time of their normally scheduled biochemical control tests.

### 2.1. Molecular Biology Assays

Peripheral blood mononuclear cells (PBMCs) from 20 mL of ethylenediamine-tetra acetic acid (EDTA) anticoagulated blood were isolated by Lympholyte-H (Cedarlane, BVT, Canada). p22^phox^ protein expression and MYPT-1 phosphorylation state were assessed by Western blot analysis as previously reported [8,9].

### 2.2. p22^phox^ Protein Expression and MYPT-1 Phosphorylation State

Total protein extract was obtained using the lysis of mononuclear cells with lysis buffer (Tris HCl 20 mmol/L, NaCl 150 mmol/L, EDTA 5.0 mmol/L, Niaproof 1.5%, Na_3_VO_4_ 1.0 mmol/L, SDS 0.1%, and PMSF 0.5 mmol/L) added to Proteases Inhibitor Cocktail (Roche Molecular Biochemicals, Mannheim, Germany) and Phosphatase Inhibitor Cocktail I (Sigma-Aldrich, St. Louis, MO, USA). Protein concentration was evaluated using a bicinchoninic acid assay (BCA Protein Assay; Pierce). The proteins were separated by SDS-PAGE in Tris pH 8.3. Protein transfer on nitrocellulose membranes was performed using a Hoefer TE 22 Mini Tank Transfer Unit (Amersham Pharmacia Biotech, Uppsala, Sweden) with the following transfer buffer: 39 mmol/L glycine, 48 mmol/L Tris base, 0.037% SDS (electrophoresis grade), and 20% methanol. The membranes were incubated overnight with primary polyclonal antibody for the detection of specific proteins: anti- p22^phox^ (Santa Cruz Biotechnologies, Santa Cruz, CA, USA), antiphospho-MYPT (Cell Signaling technology, Danvers, MA, USA), and anti-MYPT (Cell Signaling technology, Danvers, MA, USA). HRP-conjugated secondary antibodies were used (Merstham Pharmacia, Uppsala, Sweden), and immunoreactive proteins were visualized with chemiluminescence using a SuperSignal WestPico Chemiluminescent Substrate (Pierce). MYPT- 1 phosphorylation was evaluated with densitometric semiquantitative analysis using NIH imaging software. The ratio between phospho-MYPT-1 and MYPT-1 was used as an index of MYPT-1 phosphorylation.

### 2.3. HO-1 Protein Level

HO-1 protein level was measured using a commercially available enzyme-linked immunosorbent assay (ELISA) kit (IMMUNOSET^®^ HO-1 human, ELISA development set, Enzo Life Sciences, Farmingdale, NY, USA). Briefly, following the manufacturer’s instructions, 100 μg of samples was plated and incubated with the HO-1 capture antibody. After proper washing steps, a detection antibody for HO-1 was added and then incubated with streptavidin conjugated to horseradish peroxidase. Finally, after a washing step, the plate was incubated with TMB substrate, blocked with HCl 1N, and subsequently read at 450 nm using the Ensight™ Multimode Plate Reader (PerkinElmer, Waltham, MA, USA). Values are expressed as nanograms per milliliter.

### 2.4. Statistical Analysis

Statistical analysis was performed using GraphPad Prism 5 software (GraphPad Software Inc., La Jolla, CA, USA). Data are expressed as means ± standard deviations. ANOVA was used to compare quantitative variables between groups, and Student’s t-tests was used for paired variables. Values of 5% or less (*p* < 0.05) were considered significant.

The normal distribution of the data was determined using the Kolmogorov–Smirnov test.

## 3. Results

The average renal function of tolvaptan-treated ADPKD patients in terms of blood creatinine and eGFR was 126.3 ± 13.3 µmol/L and 53.8 ± 4.6 mL/min/1.73 m^2^, respectively. The blood creatinine and eGFR of untreated ADPKD patients was 78.67 ± 11.98 µmol/L and 91.44 ± 14.07 mL/min/1.73 m^2^, respectively.

### 3.1. p22^phox^ Protein Expression

As shown in Figure 1, p22^phox^ protein expression was significantly higher in untreated ADPKD patients compared to that in both tolvaptan-treated ADPKD patients and healthy subjects: 1.42 ± 0.11 vs. 0.86 ± 0.15, *p* = 0.015, vs. 0.53 ± 0.11 densitometric units, *p* < 0.001, respectively.

### 3.2. MYPT-1 Phosphorylation State

MYPT-1 phosphorylation was significantly higher in untreated ADPKD patients compared to that in both tolvaptan-treated ADPKD patients and healthy subjects: 0.96 ± 0.28 vs. 0.68 ± 0.09 densitometric units, *p* = 0.013 and vs. 0.47 ± 0.13 densitometric units, *p* < 0.001, respectively (Figure 2).

### 3.3. HO-1 Protein Level

Figure 3 shows that the HO-1 protein level was significantly higher in tolvaptan-treated ADPKD patients compared to that in both untreated ADPKD patients and healthy subjects: 5.33 ± 3.34 vs. 2.08 ± 0.79 ng/mL, *p* = 0.012, vs. 1.97 ± 1.22 ng/mL, *p* = 0.012, respectively.

## 4. Discussion

This study showed at the level of molecular biology that OxSt is activated in ADPKD, and that treatment with tolvaptan is associated not only with the reduction in proteins closely related to OxSt signaling, inflammation, and cardiovascular-renal remodeling, but also with the induction of defense/protection toward OxSt. OxSt, in terms of p22^phox^ and MYPT-1 phosphorylation state, was significantly higher in untreated ADPKD patients, whereas it was significantly reduced in tolvaptan-treated ADPKD patients. In addition, the tolvaptan-treated ADPKD patients’ HO-1 levels were significantly higher compared to those of untreated ADPKD patients.

p22^phox^ is a 22 kDa subunit of cytochrome b558 of the NADH/NADPH oxidase present both in leukocytes and in the vascular wall, and functions as an integral subunit of the final electron transport from NAD(P)H to heme and molecular oxygen in generating superoxide anions [16]. In tolvaptan-treated ADPKD patients, the reduction in p22^phox^ protein levels not only suggests reduced OxSt, but also, given its presence in leukocytes, an inhibition of leukocyte activation, a well-known cause of OxSt in CKD. As a consequence, this reduction in p22^phox^ protein levels suggests an inhibition of OxSt-mediated signaling mechanisms responsible for vascular remodeling and atherogenesis [5,6].

Another fundamental signaling pathway in OxSt processes is RhoA/ROCK [10,12]. The activation of this pathway induces OxSt via upregulation of NADPH oxidase, and the RhoA/ROCK signal is involved in the vascular effect of OxSt [10,12]. In addition, it has been recently proven that RhoA/ROCK pathway activation is promoted following the inactivation of PKD-1 via activation of proteins involved in cystogenesis [17,18], whereas ROCK inhibition reduces the activity of proteins involved in cystogenesis, which are upregulated in both PKD1-mutated cystic cells and cystic kidney tissue samples from ADPKD patients [17,18].

Another oxidative-stress-related protein evaluated in this study is heme oxygenase 1 (HO-1), the inducible isoform of HO that protects against oxidative processes. HO-1 acts on heme, producing CO and biliverdin, which is further metabolized to bilirubin, a powerful antioxidant. There are three different isoforms of HO: HO-1, HO-2, and HO-3. HO-1 has a low baseline expression that increases rapidly in the presence of oxidative stress, whereas HO-2 and HO-3 are constitutively expressed [19]. HO-1 mediates production of the CO vasodilator and contributes to the regulation of vascular tone and therefore of blood pressure and endothelial function. Finally, HO-1 has been shown to have a long-term anti-inflammatory and antiproliferative effect [16,20,21].

Tolvaptan is an arginine-vasopressin V2 receptor antagonist and is currently indicated to slow the progression of cyst development and renal failure associated with ADPKD [4]. Ishikawa et al. showed in an animal model that chronic tolvaptan treatment significantly suppressed the upregulation of NADPH oxidase subunits including p22^phox^, and inhibited RhoA expression, phosphorylation, and MYPT-1 phosphorylation [22]; Novak et al. [23] reported that vascular endothelial cell protein expression of NF-κB was increased in ADPKD, consistent with the presence of local and systemic inflammation. Furthermore, the increases in OxSt and inflammation likely contribute to the decrease in NO bioavailability, as shown in subcutaneous resistance vessels from ADPKD patients where endothelium (NO)-dependent dilatation is impaired. Finally, Fujiki et al. [24] showed that in mpkCCD cells and in the external medulla of mouse kidneys, tolvaptan increased mRNA and protein expressions of HO-1.

For the first time, the results of our study show, ex vivo in humans, that treatment with tolvaptan in ADPKD patients is associated with a reduction in OxSt and an increase in HO-1, protective against oxidative stress. Although our study was performed in a small cohort of patients, in line with the approach from molecular biology, the results support the data obtained by in vitro and animal model studies and provide the mechanistic rationale for the reductions in OxSt and inflammation, as well as protection against the worsening of renal function, endothelial damage, and hypertension in tolvaptan-treated ADPKD patients.

A relevant limitation of our study should be mentioned, which is the small number of selected patients. However, on the one hand, to the best of our knowledge, as this is the first study in humans using a molecular biology approach to deeply investigate the status of OxSt in ADPKD patients, the impact of tolvaptan treatment on OxSt may justify the limited number of patients enrolled; on the other hand, the results of this study may serve as a useful working hypothesis for further studies with a larger number of patients enrolled or an extended study duration to allow the benefits of tolvaptan treatment on OxSt/inflammation-related worsening of renal function in ADPKD to be conclusively demonstrated.

## Figures and Tables

**Figure 1 jcm-11-00402-f001:**
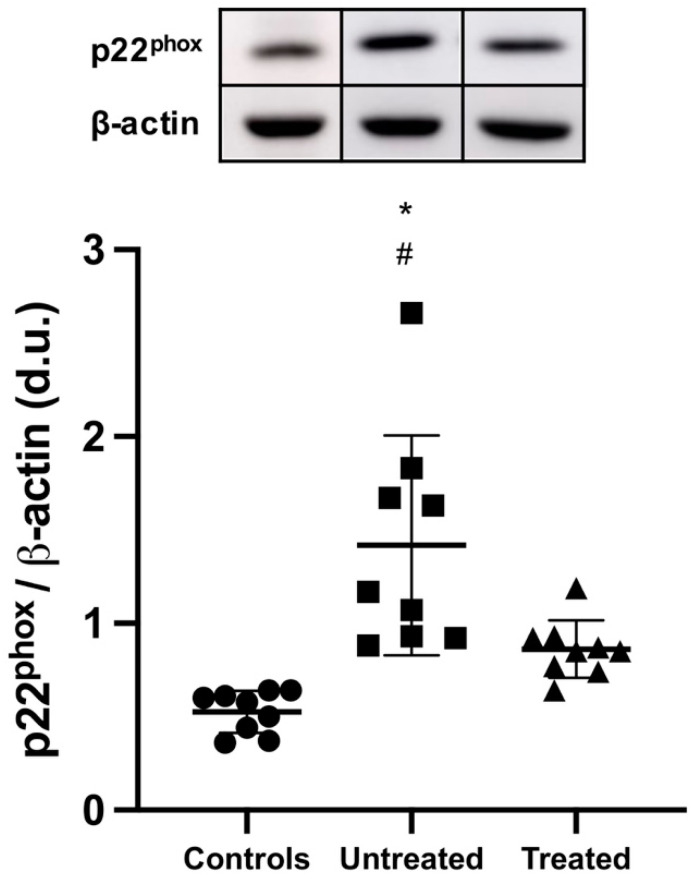
p22^phox^ protein expression. A representative Western blot of protein lysates harvested from controls, untreated ADPKD patients, and tolvaptan-treated ADPKD patients. Compared to controls, p22^phox^ expression was significantly increased in untreated ADPKD (*: *p* < 0.001) and in treated ADPKD patients (#: *p* = 0.015).

**Figure 2 jcm-11-00402-f002:**
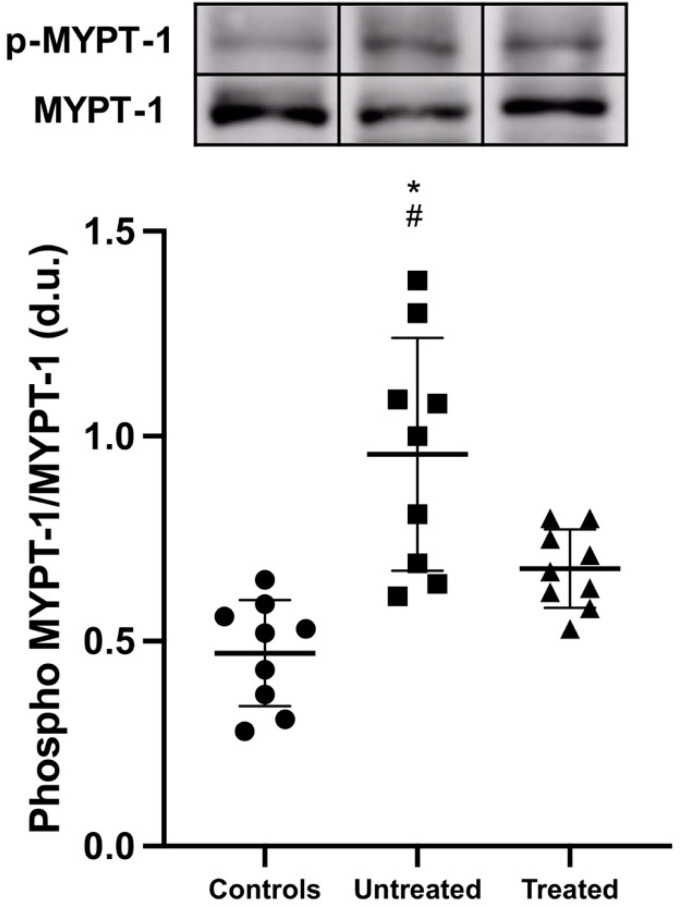
MYPT-1 phosphorylation state. A representative Western blot of protein lysates harvested from controls, untreated ADPKD patients and tolvaptan-treated ADPKD patients. Compared to controls, phospho-MYPT-1 was significantly increased in untreated ADPKD patients (*: *p* < 0.001) and in treated ADPKD patients (#: *p* = 0.013).

**Figure 3 jcm-11-00402-f003:**
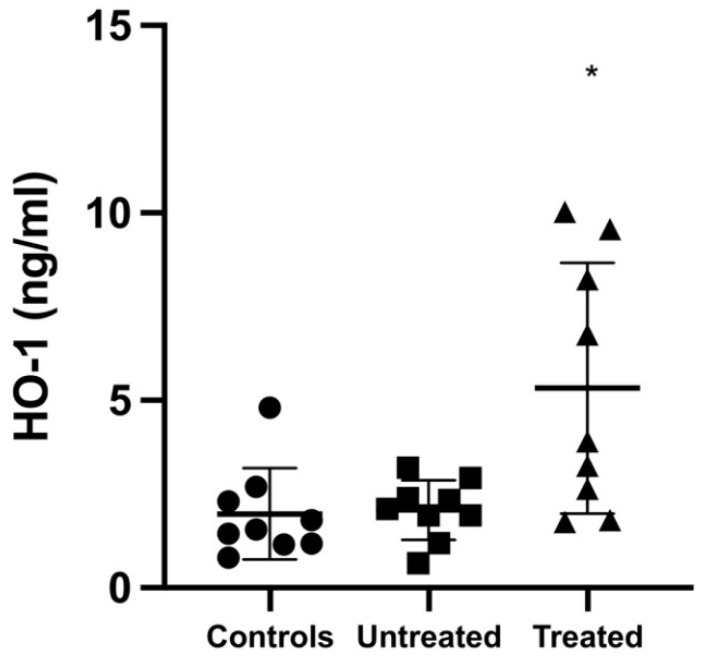
HO-1 protein expression. The HO-1 protein concentration was significantly higher in ADPKD patients treated with tolvaptan compared to that in both untreated ADPKD patients and controls (*: *p* = 0.012). Data are expressed as mean ± SD.

## Data Availability

The data presented in this study are available on request from the corresponding author.

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
