# Peer review of "Effects of Tolvaptan on Oxidative Stress in ADPKD: A Molecular Biological Approach"

_jcm, 2022, doi:10.3390/jcm11020402_

Round 1

Reviewer 1 Report

Dear authors,

With interest, I read your article on the effect of tolvaptan on oxidative stress in ADPKD. Tolvaptan is so far the only available treatment option in this disease and molecular mechanism is not established yet. It is for instance unknown why, after longer time use, eGFR decline is attenuated, while effects on total kidney volume are decreased.

Your article is clear and well written. I have however 3 major remarks that could be improved:

  1. numbers are low, you measured only 9 ADPKD patients
  2. there are large differences in kidney function and age between the treated and control group, would be good to comment on this in the discussion
  3. you now look at oxidative stress at mRNA level, however, it would definitely strengthen your findings and make them of more importance if you could also show systemic effects of oxidative stress, by measuring for instance nitrate/free thiols/MDA TBARS/glutathion (GSS)/ oxidated glutathon (GSSG) in blood.

Reviewer 2 Report

Abstract:  Ideally you should indicate for how you controlled for various factors that affect oxidative stress like smoking, age, lifestyle, but I understand there are space constraints.

Introduction:  This is rather long.  It could be improved by shortening and cutting back on the points not related to the main objective of this study.  --

The point about oxidative stress being evaluated in vitro and in animal models of ADPKD should be referenced.

Patients and Methods:  There is quite a large age gap between the patients on versus not on Tolvaptam as well as controls.  It would be useful to have a table showing how these three groups (ADPKD on tolvaptom, ADPKD not on Tolvaptam and healthy subjects) compare in terms of age, gender, smoking, eGFR, height adjust total kidney volumes, liver volumes and other manifestations of ADPKD as well as other factors known to contribute to oxidatative stress.

Statistical analysis:  Indicate how you determined if the data were distributed normally.  Indicate how you controlled for possible confounding variables.

Figures: good

Discussion  There needs to be a limitations section that addresses that possibility of confounding variables.

It would be useful to enroll more subjects if this is possible.
